# Atraric Acid Ameliorates Hyperpigmentation through the Downregulation of the PKA/CREB/MITF Signaling Pathway

**DOI:** 10.3390/ijms232415952

**Published:** 2022-12-15

**Authors:** Jing Li, Shengping Jiang, Chengyun Huang, Xiaolong Yang

**Affiliations:** School of Pharmaceutical Sciences, South-Central Minzu University, Wuhan 430079, China

**Keywords:** atraric acid, melanin, hyperpigmentation, MITF, PKA/CREB/MITF signaling pathway

## Abstract

Atraric acid (AA) is derived from lichens and is widely used in perfumes for its desirable scent. It has been reported as having anti-inflammatory and antioxidant activity. Hyperpigmentation is the underlying cause of a variety of dermatological diseases that have a significant impact on patients’ quality of life and are frequently difficult to treat. This study aimed to explore the inhibitory effects of AA on hyperpigmentation in vitro and in vivo and its potential molecular mechanisms. The cytological results revealed that at a dose of 250 μM, AA may reduce melanin content and tyrosinase levels without causing cytotoxicity. Furthermore, the expression of melanocortin-1 receptor (MC1R), phosphorylated protein kinase A (pPKA) and phosphorylated cAMP response element binding protein (pCREB) were downregulated in AA-administrated cells. In vivo, histological analysis showed that AA could inhibit melanin production and tyrosinase activity, and 3% AA had the best activity, with almost no side effects. Furthermore, the results of Western blot analysis and RT-PCR suggested that AA may suppress the mRNA transcription of microphthalmia-associated transcription factor (MITF) protein and tyrosine protease by decreasing the expression of MC1R, consequently decreasing the phosphorylation of PKA and CREB. Finally, the MC1R inhibitor MSG606 verified the hypothesis that AA suppresses melanin formation by downregulating the PKA/CREB/MITF signaling pathway. Taken together, our study offers valuable information for the development of AA as a possible ingredient in skin-lightening cosmeceuticals and hyperpigmentation inhibitors.

## 1. Introduction

Melanin is secreted by melanocytes located in the basal layer of the epidermis and is an essential element of skin, hair and eye color. Melanin is produced through a complex series of enzymatic and chemical reactions involved in melanogenesis. The tyrosinase enzyme plays a key role in melanogenesis [1,2]. MITF is an important biomarker that controls the transcription of critical melanogenic proteins [3]. It is widely accepted that several signaling pathways are involved in α-melanocyte-stimulating hormone (α-MSH)-induced melanogenesis by regulating MITF, including those mediated by PKA/CREB, mitogen-activated protein kinase (MAPK)/extracellular signal-regulated kinase (ERK), nuclear factor kappa-B (NF-κB)/c-Jun N-terminal kinase (JNK) and human phosphoinositide-3 kinase (PI3K)/protein kinase B (AKT) [4,5]. Once the MC1R on melanocytes is stimulated by α-MSH, it activates the production of cAMP by activating adenylate cyclase. Further activation of PKA by cAMP, which phosphorylates the CREB, results in MITF up-regulation. The inhibition of MAPK kinase inhibits MITF turnover in the presence of cycloheximide, thus implicating ERK/MAPK signaling in MITF degradation [6]. In addition, the melanin inhibitors of IL-1β largely inhibit the expression of MITF by the NF-κB/JNK pathway, thereby inhibiting melanogenesis [7]. Some compounds, such as gallic acid, inhibit hyperpigmentation via the PI3K/AKT signaling pathway [8]. Under normal physiological conditions, melanin formation is critical for protecting DNA and skin from UV radiation. However, hyperpigmentation, or excessive melanin production, is the underlying cause of a variety of dermatological diseases, such as melasma, freckles, age spots and malignant melanoma, which have a significant impact on patients’ quality of life and are frequently difficult to cure [9]. Thus, the development of safer, more robust, and efficient skin protective effects is essential for preventing and blocking diseases associated with UVB-mediated skin hyperpigmentation.

Currently, kojic acid and arbutin are well-known melanogenesis inhibitors. However, the long-term use of these agents may not only cause various safety concerns, but also achieve unsatisfactory efficacy in most cases [10,11]. Lasers and light sources, which have good aesthetic advantages but a higher risk of adverse effects, have also been recommended to improve this condition [12]. On the basis of safety concerns, numerous researchers have attempted to identify melanin inhibitors derived from natural products.

Lichens are derived from the symbiotic association of a fungus with a green alga or cyanobacterium, which is widely used to treat skin disorders, diabetes and hypertension in traditional medicine [13]. AA is derived from lichens and is widely used in perfumes for its desirable scent. AA has a long history of safety as a substance that can touch human skin and has shown anti-inflammatory, antioxidant and antibacterial activities [14,15,16]. However, whether AA inhibits hyperpigmentation has not been reported in the United States or elsewhere. As a result, the purpose of this study is to discover the effect of AA on melanin inhibition and its likely mechanism.

## 2. Results

### 2.1. Atraric Acid Has No Cytotoxicity in α-MSH-Induced PIG1 Cells

We first aimed to evaluate the cytotoxic effect and determine the maximum concentration to apply to future experiments of AA in PIG1 cells by using a Cell Counting Kit-8 (CCK8) assay. After treatment with AA for 24 h or 48 h, cell viability was still above 80% with 250 µM of AA in PIG1 (Figure 1b) cells. The International Organization for Standardization (ISO) 10993–5:2009 (Biological Evaluation of Medical Devices) defines non-cytotoxicity as cell viability higher than 80% [16]. Therefore, a concentration of AA at 250 µM was selected for the following experiments of atraric acid on PIG1 cells. Cells were treated with AA for 24 h and 48 h, and then cell viability was determined by the CCK8 assay (b). Data are expressed as the mean ± SEM from three independent experiments (*n* = 3 per group).

### 2.2. AA Suppresses Melanin Synthesis and Tyrosinase in α-MSH-Induced PIG1 Cells

The anti-melanogenic effect of AA was evaluated by determining the melanin content and tyrosinase activity in PIG1 cells after treatment with α-MSH and AA/arbutin versus a vehicle control for 24 h. Immunofluorescence results showed that the fluorescence expression of α-MSH in the control group was significantly lower than that in the other three groups (Appendix A). The result indicated that α-MSH-induced melanogenesis in the PIG1 cells model was successful. As shown in Figure 2a, AA could significantly (*p* < 0.01) inhibit α-MSH-induced melanin content. Notably, AA was more effective than arbutin in inhibiting melanin production. Because tyrosinase is a key melanogenic enzyme that controls melanin production [17], we also determined the effects of AA and arbutin on tyrosinase activity in PIG1 cells under the aforementioned conditions. The results indicated that AA could substantially (*p* < 0.001) suppress α-MSH-induced tyrosinase activity, paralleling the inhibitory effect on melanin production (Figure 2b).

### 2.3. Molecular Docking of Relation Proteins of Melanogenesis

Several signaling pathways have been implicated in α-MSH-induced melanogenesis through the regulation of MITF, including those mediated by PKA/CREB, MAPK/ERK, MAPK, JNK, and PI3K/AKT [4,5]. Therefore, with the use of molecular docking, we carried out an in-depth investigation into the effects that AA has on the expression of possible upstream mediators. The binding energy values of AA and relation proteins obtained from the DS 3.0 binding energy program are shown in Table 1. The interactions between AA and PKA and CREB were shown to be more stable and powerful than those between AA and other related proteins, with binding energies of −12.28 Kcal/mol and −23.005 Kcal/mol, respectively (Figure 3a,b).

The phosphorylation of tyrosine and serine/threonine residues is the most prevalent post-translational modification of proteins [18]. As shown in Figure 3, the interaction between amino acid residues at the PKA binding site and AA were shown as follows: TYR59, THR75 and GLN74 (carbon hydrogen bond); GLN132 and GLN39 (conventional hydrogen bond); VAL54 (pi-alkyl and alkyl); VAL76 and LEU121 (alkyl); MET123 (pi-sulfur). The interactions between amino acid residues at the CREB binding site and AA were as follows: TYR87 and SER9 (conventional hydrogen bond), GLY101 (carbon hydrogen bond), LYS103 (attractive charge, pi-cation and unfavorable donor-donor). The result suggested that AA can bind to tyrosine, threonine or serine residues of PKA and CREB proteins. Therefore, we hypothesized that AA was able to inhibit melanin pigmentation by promoting PKA and CREB phosphorylation.

### 2.4. Effects of AA on the Expression Levels of Key Mediators of Melanogenesis in α-MSH-Induced PIG1 Cells

To explore the molecular mechanisms underlying the anti-melanogenic effect of AA, we examined the effects of AA on the protein expression levels of MC1R and MITF in α-MSH-induced PIG1 cells. MC1R is found on the surface of melanocytes [19]. Activation of MC1R sets in motion a signaling cascade through MITF, which leads to the transcription of the genes for melanogenesis [20]. While α-MSH stimulated an increase in the expression of MC1R and MITF, AA significantly reduced the expression of those two proteins, *p* < 0.01 (Figure 4a–c). 

### 2.5. Effects of AA on the Phosphorylation of CREB and PKA in α-MSH-Induced PIG1 Cells

The PKA/CREB signaling pathway is a key pathway regulating MITF to increase melanin production. According to the results of molecular docking, AA had binding energies of −12.28 Kcal/mol and −23.005 Kcal/mol, respectively. Therefore, to further understand the mechanisms involved in the regulation of MITF expression by AA, we next investigated the phosphorylation of PKA and CREB. Judging from Figure 4a,d,e, α-MSH significantly (*p* < 0.01) elevated the phosphorylation of PKA and CREB as compared to the control group. Moreover, an appreciable increase in 3% HQ, DM and HM groups in PKA and CREB phosphorylation levels was observed in α-MSH-treated PIG1 cells. These results suggested that AA may suppress the expression of MITF by reducing MC1R protein expression on the melanocytes’ surface, thereby down-regulating the phosphorylation of PKA and CREB, leading to a decrease in melanogenesis.

### 2.6. Effects of AA on UVB-Induced Hyperpigmentation in Guinea Pigs 

α-MSH and UVB irradiation can stimulate melanin secretion, thereby triggering melanogenesis. Therefore, to observe the effects of AA on UVB-induced hyperpigmentation in guinea pigs, H&E staining tests were conducted. As shown in Figure 5, histological analysis confirmed that AA on the DM and HM groups and 3% HQ markedly reduced pigmentation in the superficial layer of the skin (green arrows). However, it is worth noting there was an epidermal hyperplasia on the skin surface of the 3% HQ. All in all, the results above indicated that AA could effectively inhibit UVB-induced pigmentation, and the effect was better than 3% HQ.

However, in the pathological sections, we found a change in epidermal hyperplasia in the guinea pigs in the high dose group. Therefore, we further evaluated the liver index, kidney index, spleen index, and pathological sections of the liver and kidney in the guinea pig. The results showed that the liver index and the pathological sections of the liver and kidney were abnormal in the positive drug group and the high-dose group guinea pigs (Appendix A). In summary, we concluded that 3% AA (MD group) had the best effect for inhibiting UVB-induced hyperpigmentation.

However, in the pathological sections, we found a change in epidermal hyperplasia in the guinea pigs in the high dose group. Therefore, we further evaluated the liver index, kidney index, and spleen index in the guinea pig. As shown in Appendix A, the spleen index of all groups in guinea pigs was almost the same. However, there were effects of different drugs on the liver and kidney indices in guinea pigs compared with the control group. Therefore, we further carried out a pathological section study on the liver and kidney. The results showed that the guinea pig liver tissue showed blurred cell boundaries and vacuolization in the model group, the positive drug group and the high-dose group (Appendix A). Besides, the kidney pathological sections (Appendix A) in the model group, the positive drug group and the high-dose group showed serious renal fibrosis (fibrosis appeared blue). In summary, we concluded that 3% AA (MD group) had the best effect in inhibiting UVB-induced hyperpigmentation.

### 2.7. Effects of AA on Expression Levels of Melanogenesis-Related Genes In Vivo

In order to further explore the molecular mechanism underlying the AA inhibition of melanin production in vivo, the expression levels of MC1R, MITF, pCREB and pPKA proteins were determined using Western blot. As demonstrated in Figure 6, compared with the model group, the protein expressions of MC1R, MITF, pCREB and pPKA in the skin of the 3% HQ group, DM group and HM group were significantly decreased (Figure 6b–e, *p* < 0.05). These results were consistent with the results of the PIG1 cell experiments.

Then, using RT-PCR, we investigated the effects of AA on the relative quantification of the mRNA transcriptions of the MITF and tyrosinase genes. According to Figure 7, the mRNA levels of the MITF protein and tyrosinase protein were decreased by the AA. Taken together, the AA suppression of melanogenesis may be via inhibiting the PKA/CREB/MITF pathway and decreasing the expression of the MITF and tyrosinase genes.

### 2.8. Effects of AA on the PKA/CREB/MITF Signaling Pathway

The MC1R inhibitor MSG606 was utilized to study the participation signaling pathway in AA-mediated antimelanogenesis to confirm whether the PKA/CREB/MITF pathway is implicated in the inhibition of melanogenesis by AA. We measured the melanin content in the PIG1 cells administration with α-MSH, AA and MSG606 for 24 h. As shown in Figure 8c, the PIG1 cells in the MSG606 + α-MSH group, MSG606 + α-MSH + AA group and MSG606 + AA group showed significantly lower (*p* < 0.01) levels of melanin in comparison to the α-MSH group. Noteworthy, co-treatment with α-MSH and AA reduced melanin content compared with that in the α-MSH + MSG606 group. The expression levels of pPKA in the different groups were consistent with the contents of melanin in the different groups (Figure 8a,b).

In addition, the forskolin (FSK) and IBMX were shown to be able to increase melanin content via the cAMP/PKA/CREB pathway [21]. Therefore, we measured the melanin content and the protein expression levels of MC1R, MITF, pCREB and pPKA in the PIG1 cells administration with FSK and IBMX. As shown in Figure 8d–h, compared with the control group, FSK and IBMX stimulated the protein expressions of MC1R, MITF, pCREB and pPKA in the PIG1 cells. However, the expression levels of MC1R, MITF, pCREB and pPKA were significantly reduced by the AA of the FSK group and IBMX group. In addition, we measured the melanin content in cells of the different groups. The results were consistent with the results of the protein expressions (Figure 8i). The results above indicated that the pharmaceutical mechanism of AA could belong to the PKA/CREB/MITF pathway. 

## 3. Discussion

Melanosomes are transported to neighboring keratinocytes and the upper epidermis for DNA photoprotection after UV exposure. To limit cell development with unrepaired DNA damage, they induce apoptosis in melanin-containing keratinocytes in the upper epidermis. Keratinocytes further release several growth factors, such as α-MSH, and aid in UV-induced hyperpigmentation [22]. In addition, tyrosinase plays a vital role as the rate-limiting enzyme that manages melanin production. The increase in melanin production is directly related to the activity of tyrosinase [23]. Therefore, we investigated whether AA could inhibit hyperpigmentation through the inhibition of tyrosinase activity in α-MSH-induced PIG1 cells and in UVB-induced guinea pigs. 

We discovered that AA significantly inhibited melanin synthesis and tyrosinase activity at a dose of 250 μM and had no cytotoxic effect on PIG1cells. Furthermore, in the UVB-induced hyperpigmentation guinea pig model, we also found that AA could inhibit melanin production and tyrosinase activity, and 3% AA (MD group) had the best activity, with almost no side effects. 

The cutaneous response to UVB, mediated by the MC1R, affects the expression of melanin-associated genes [24]. Activation of MC1R initiates a signaling cascade through cAMP that leads to the transcription of melanogenesis genes. First, the key transcription factor MITF is activated. Immediately after, MITF initiates the transcription of downstream tyrosinase [20]. In this study, we confirmed the role of AA in inhibiting melanin production via the regulation of MITF using the α-MSH-induced PIG1 cells in vitro and UVB-induced guinea pig model in vivo.

Activation of the PKA/CREB signaling pathway is known to play an essential role in MITF expression and activity. Phosphorylation of CREB is activated by the phosphorylation of PKA, leading to the expression of MITF and resulting in an increase in melanin synthesis [3,25]. Thus, the downregulation of PKA and CREB signaling in melanocytes inhibits melanin synthesis by downregulating MITF expression. Lee et al. [26] proved that protocatechuic aldehyde decreased the expression levels of MITF by inhibiting the phosphorylation of CREB and PKA in response to cAMP. Chu et al. [27] demonstrated that calycosin inhibited melanin synthesis by ameliorating the PKA/CREB and p38 MAPK signaling pathways. Consistent with these findings, AA might reduce the expression of MC1R, so decreasing the phosphorylation of PKA and CREB, which could limit the mRNA transcription of MITF protein and tyrosine protease, thereby preventing hyperpigmentation.

In addition, in the α-MSH-induced PIG1 cell model, we knocked down the MC1R and administrated it with AA, and found that AA could reduce the melanin content and the level of pPKA more than the α-MSH-induced PIG1 cells administrated with the MC1R inhibitor MSG606 (Figure 8a,b). Therefore, AA might be able to regulate both MC1R and pPKA. In addition, we found that the protein expression levels of MC1R, MITF, pPKA and pCREB, and the melanin content in FSK- or IBMX-induced PIG1 cells, were reduced by AA. In conclusion, all data in this study suggest that the administration of AA effectively prevented hyperpigmentation associated with elevated α-MSH, which might be mediated via the regulation of the PKA/CREB/MITF signaling pathway. 

## 4. Materials and Methods

### 4.1. Materials

Atraric acid was synthesized by our laboratory (purity ≥ 97%, Figure 1a). MSG606 was purchased from MCE Med Chem Express (Shanghai, China). Arbutin and α-MSH were purchased from Sigma Aldrich (St. Louis, MO, USA). CCK8 kits and ELISA kits were obtained from Biosharp (Ranjek Technology Co., LTD, Anhui, China). Antibodies used in this study and their sources were as follows: β-actin, MC1R, MITF, tyrosinase, and β-catenin (Bioswamp, Wuhan, China).

### 4.2. Preparation of Atraric Acid 

An amount of 10 g AA was synthesized according to our laboratory’s previously described method [28].

### 4.3. Molecular Docking Analysis

We used Chem3D to draw AA’s 3D structure and imported it into Discovery Studio (2019) for small molecule preprocessing. Meanwhile, we downloaded the proteins CERB (PDB ID: 7KFO), PKA (PDB ID: 4YKO), MAPK (PDB ID: 6XQ9), AKT (PDB ID: 7D85), PI3K (PDB ID: 7CIO), EKR (PDB ID: 4QTA) and JNK (PDB ID: 6LOT) from the PDB database and imported them into Discovery Studio (2019). The imported proteins were processed separately to remove water molecules and unwanted structures, followed by hydroprocessing, protein modification and docking site selection. After processing the small molecule and protein in AA, semi-flexible molecule docking between the protein and AA was performed. The connection with the highest score was selected for analysis.

### 4.4. Cell Culture and Cell Viability Assay

The PIG1 cells were purchased from the China Centre for Type Culture Collection (Wuhan, China). The PIG1 cells were cultured in Dulbecco’s modified Eagle’s medium (Gbico, Carlsbad, CA, USA) supplemented with 10% (*v*/*v*) fetal bovine serum (Gbico, Carlsbad, CA, USA), and 100 U/mL penicillin–streptomycin (P/S) (Tianhang, Shanghai, China). The PIG1 cells were sub-cultured every three days and maintained at 37 °C with 5% CO_2_ in a humidified incubator.

AA was dissolved in dimethyl sulfoxide (DMSO) to make a 1 M concentration stock solution and stored at 4 ℃. Before use, this solution was diluted to the desired concentration with the culture solution. Briefly, the PIG1 cells were inoculated into the 96-well culture plates (1 × 10^5^ cells/well). Then, the cells were treated with corresponding drugs for different exposure durations according to the study design. Cells were exposed to different concentrations of AA (62.5, 125, 250, 500 and 1000 μM) for 24 h or 48 h, respectively. Then, the cell survival rate was measured by the CCK8 assay.

An amount of 0.25% Trypsin-EDTA (1×) (Gbico, Carlsbad, CA, USA) was added to the solution during the logarithmic phase of adherence to the PIG1 cells. Then, 100 μL of the suspension was added to each hole of the 96-well culture plate and cultured for 24 h. The PIG1 cells were categorized into 4 groups and treated with α-MSH (100 nM), α-MSH + arbutin (500 µM), and α-MSH + AA (250 µM) for 24 h after removing the culture solution, with the culture solution serving as the control.

Cell viabilities were expressed as percentages of viable cells relative to the corresponding vehicle-treated control group.

### 4.5. Animals

All male guinea pigs (300–450 g) were bought from the Institutional Animal Care and Use Committee (IACUC), Wuhan, China (AUP No.42817300001954). All guinea pigs were housed in a temperature-controlled room (23 ℃ ± 24 ℃) with food and water ad libitum and maintained on a 12 h light-dark cycle. 

A total of 48 guinea pigs were randomly divided into 6 groups with 8 animals per group as follows: control group (C); model group (M, UVB irradiation); positive group (P, 3% hydroquinone cream, HQ); low dose (LD, 1% AA); medium dose (MD, 3% AA); high dose (HD, 6% AA). An amount of 0.1g of AA was weighed and dissolved into 10 mL of normal saline to prepare 1% AA. An amount of 3% AA and 6% AA were configured according to the same method. UVB-induced hyperpigmentation was induced on the backs of the guinea pigs. UVB irradiation conditions were the same as previous studies with a slight modification [29,30]. Briefly, the separate areas (2.5 cm × 2.5 cm) of the back of each guinea pig were exposed to the UVB radiation (Waldman UV 800, Herbert Waldmann GmbH, Philips TL/12 lamp emitting 280–350 nm). The total UVB dose was 500 mj/cm^2^ per exposure. The dorsal skin of each guinea pig was painted separately with 3% hydroquinone cream or AA or saline at 200 μL before UVB irradiation. The experiment was conducted for 8 weeks, and skin, liver, kidney, and spleen tissue samples from the guinea pigs were collected for analysis.

### 4.6. Melanin Contents and Tyrosinase Activity Assay

Melanin contents in the cells or skin of guinea pigs were measured by Enzyme-Linked Immunosorbent Assay (ELISA) test kits according to the manufacturer’s protocol.

Tyrosinase activity was measured using a previously described method [26] with minor modifications. Briefly, PIG1cells were stimulated with α-MSH and treated with various concentrations of AA and arbutin. After incubation at 37 °C for 24 h, the cells were washed with PBS and lysed in 50 nM sodium phosphate buffer (pH 6.8) containing 1% Triton X-100 and 0.1 mM phenylmethylsulfonyl fluoride (PMFS). Cell lysates were harvested by centrifugation at 12,000 rpm for 15 min at 4 ℃. After quantification and normalization, the cell lysate was incubated with the freshly prepared 15 mM L-DOPA at 37 ℃ for 1 h to promote dopachrome formation. The solution was then measured using a spectrophotometer at 490 nm.

### 4.7. Immunofluorescence Staining

Adherent PIG1 cells were collected and those cells (2 × 10^5^/well) were seeded on a chamber side (Nunc Lab Tek, Thermo Fisher Scientific, Waltham, MA, USA) in DMEM either with or without AA. After 48 h, the cells were fixed with 4% paraformaldehyde for 10 min, rinsed with PBS for 10 min, treated with 5% Triton X-100 for 10 min and, thereafter, incubated for 1 h with BSA/PBS. After rinsing, the cells were incubated at −4 ℃ overnight with a primary antibody (anti-human antibody against α-MSH, dilution 1:300, Google). The next day, secondary antibodies (goat anti-mouse IgGDy Light 488) were added to the tissues, followed by incubation in the dark for 50 min and nuclear staining angina by DAPI. Images were collected and analyzed under a fluorescence microscope (DM IL LED, Leica), and cell fluorescence was measured using Image J software (ver. 1.8.0).

### 4.8. Histology Analyses of the Skin, Liver and Kidney

To check the change in skin, liver and kidney, hematoxylin and eosin (H&E) and Masson staining were conducted. The thoracic aorta, liver and kidney were fixed with 10% (*v*/*v*) neutral formalin for 48 h. Those tissues were paraffin-embedded, sectioned at 4 μm, and stained with H&E and Masson to observe the change.

The experimental procedures were determined using a previously described method [31]. Tissue sections were dewaxed in xylene, rehydrated in decreasing concentrations of ethanol, washed in phosphate-buffered saline (PBS) and stained with H&E and Masson’s trichrome. After staining, each slice was observed for pathological changes under a microscope and photographs of each of the slides were taken at 100×, 200× or 400× magnification.

### 4.9. Western Blot Analysis

The PIG1 cells were seeded at a density of 6 × 10^5^ cells/mL. After incubation overnight, α-MSH (100 nM), AA (250 µM) or arbutin (500 µM) were added and further incubated for 24 h. Protein preparation and whole cell lysates were performed as described previously [31]. Specific antibodies were used to detect the total form of MCIR, MITF, tyrosinase and β-actin, which were visualized with chemiluminescence reagents [27]. The Western blot analysis of skin tissue was consistent with the above procedure.

### 4.10. Quantitative Reverse Transcription Polymerase Chain Reaction (RT-qPCR)

Total RNA was isolated and reverse transcribed to cDNA using an RNeasy mini kit (TrasGen Biotech, Beijing, China) and TOOLS Easy Fast RT kit (Solarbio, Beijing, China), respectively, according to the manufacturer’s instructions. qRT-PCR was carried out using the StepOne system with Fast SYBR Green Master Mix (Thermo Fisher Scientific) and the following primers: MITF: 5′-CCTGGAAAACCCCACCAAGT-3′ and 5′-ATGCACGAAGCTCGAGAGTG-3′; Tyrosinase: 5′-AGTAGCATGCACAACGCTCT-3′ and 5′ -TAGGTGCATTGGCTTCTGGG-3′. A total of five RNA/cDNA skin samples were used for all genes tested in triplicate. The relative expression of each target gene was determined using the 2^−∆∆CT^ method using the housekeeping gene β-actin as the reference control.

### 4.11. Statistical Analysis

Data were expressed as means ± standard error of mean (SEM). Differences between the two groups were compared using the Student’s *t*-test, and one-way ANOVA, with Bonferroni multiple comparison tests used for three or more groups. The statistical tests were performed using GraphPad Prism 5.0 software. Two-tailed *p* < 0.05 was considered statistically significant (α = 0.05).

## 5. Conclusions

For a long time, the discovery of UVB melanin inhibitors has been an attractive goal not only for whitening and lightening spots, but also for treating skin cancer. The results obtained in this study indicated for the first time that AA inhibits not only in vitro melanin in α-MSH-induced hyperpigmentation but also in vivo melanin in UVB-induced hyperpigmentation. Additionally, we found that AA exerts favorable inhibition effects on UVB-inducted hyperpigmentation, which might be mediated via the PKA/CREB/MITF signaling pathway to reduce melanin synthesis. Moreover, co-treatment with α-MSH and AA reduced the melanin content and the level of pAKT compared with that in the α-MSH + MSG606 group, which suggested that AA was probably ameliorating the PKA/CREB/MITF pathway by regulating both MC1R and PKA protein targets. However the identification of the protein targets of MCIR and PKA requires further experimental studies. Moreover, more extensive and further studies are needed for a thorough understanding of the potential inhibition of AA in melanogenesis. In summary, our findings have provided valuable information to aid the development of AA as a potential skin-lightening cosmeceutical or hyperpigmentation inhibitor. 

## Figures and Tables

**Figure 1 ijms-23-15952-f001:**
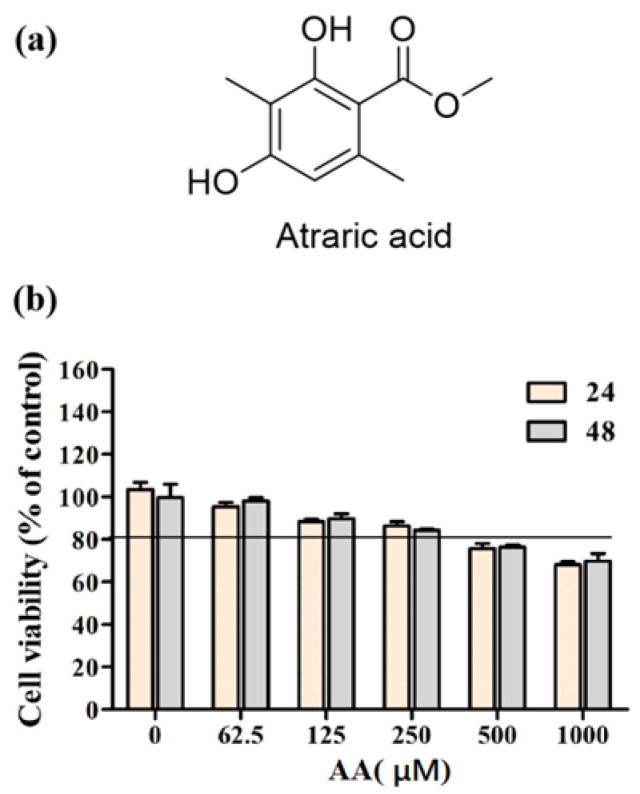
Chemical structure of atraric acid (**a**). The viability of different concentrations (**b**).

**Figure 2 ijms-23-15952-f002:**
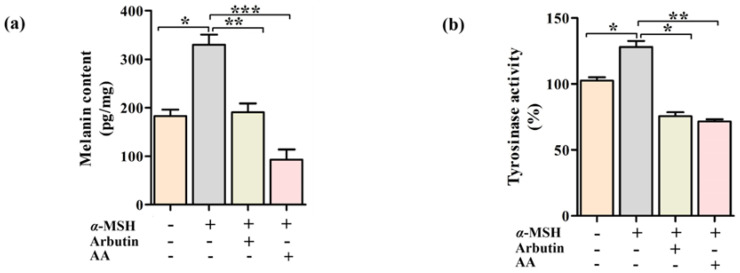
Effects of AA on cellular melanin synthesis and intracellular tyrosinase activity in α-MSH-induced PIG1 cells. Cells were treated with the indicated concentrations of AA for 24 h. The relative melanin content (**a**). The tyrosinase activity (**b**). Data are expressed as the mean ± SEM. * *p* < 0.05, ** *p* < 0.01 and *** *p* < 0.001 (*n* = 3 per group).

**Figure 3 ijms-23-15952-f003:**
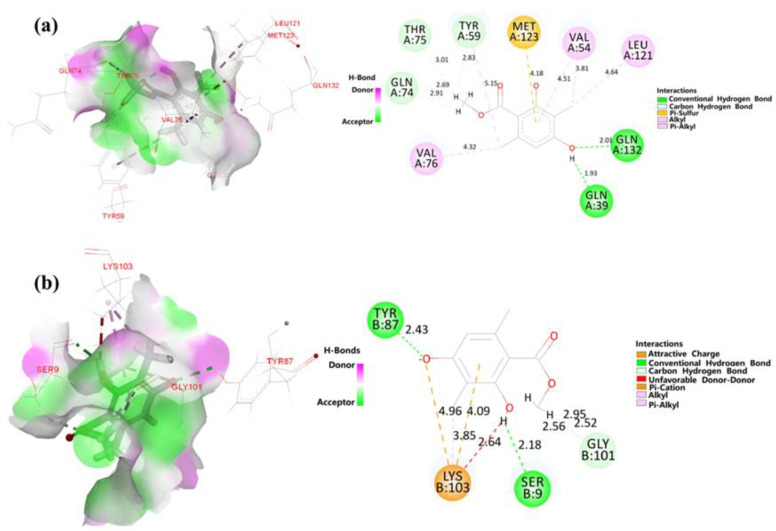
The specific interactions between AA and tyrosinase after automated docking of AA to the tyrosinase enzyme binding site. Predicted 3D structure of the PKA (Protein Data Bank; PDB ID: 4YKO)-AA complex and 2D diagram (**a**). Predicted 3D structure of the CREB (PDB ID: 7KFO)-AA complex and 2D diagram (**b**).

**Figure 4 ijms-23-15952-f004:**
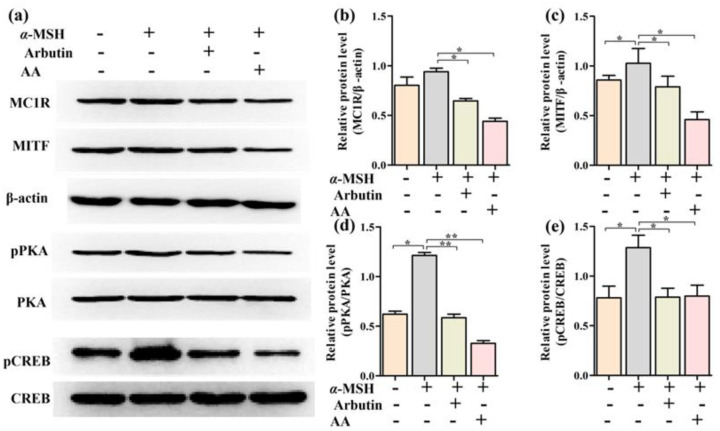
Effects of AA on the expression of MC1R, MITF, pPKA and pCREB of α-MSH-induced PIG1 cells (**a**). The levels of MC1R (**b**), MITF (**c**), pPKA (**d**) and pCREB (**e**) proteins were quantified by densitometry. Data are expressed as mean ± SEM. * *p* < 0.05, ** *p* < 0.01 (*n* = 3 per group).

**Figure 5 ijms-23-15952-f005:**
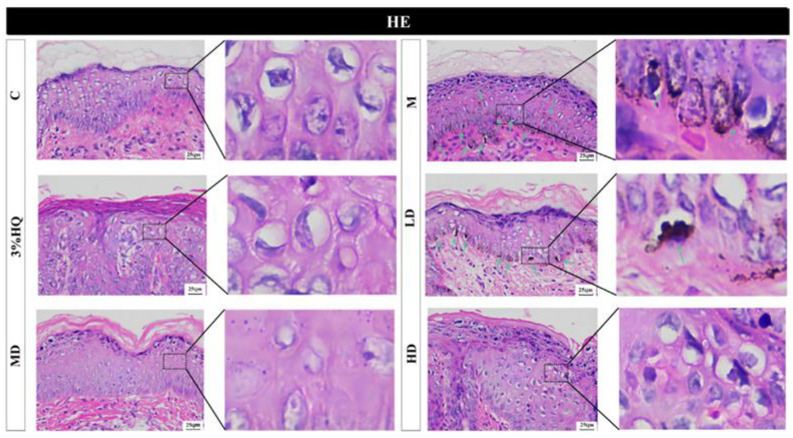
Effects of AA on histopathological features of dorsal skin. H&E staining green arrow indicates the melanin. C mean control group, M mean model group, 3% HQ mean 3% hydroquinone cream group, LD mean low dose group, MD mean medium dose group, and HD mean high dose group (*n* = 6 per group).

**Figure 6 ijms-23-15952-f006:**
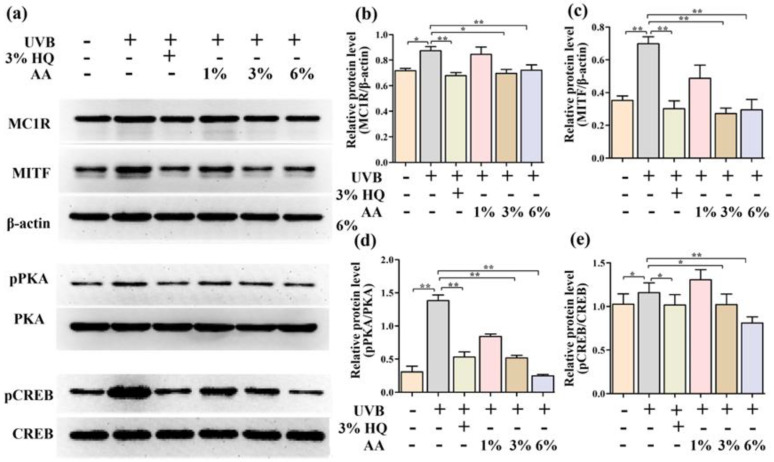
Effects of AA on the expression of MC1R, MITF, pPKA and pCREB of dorsal skin (**a**). The levels of MC1R (**b**), MITF (**c**), pPKA (**d**) and pCREB (**e**) proteins were quantified by densitometry. Data are expressed as mean ± SEM. * *p* < 0.05, ** *p* < 0.01 (*n* = 6 per group).

**Figure 7 ijms-23-15952-f007:**
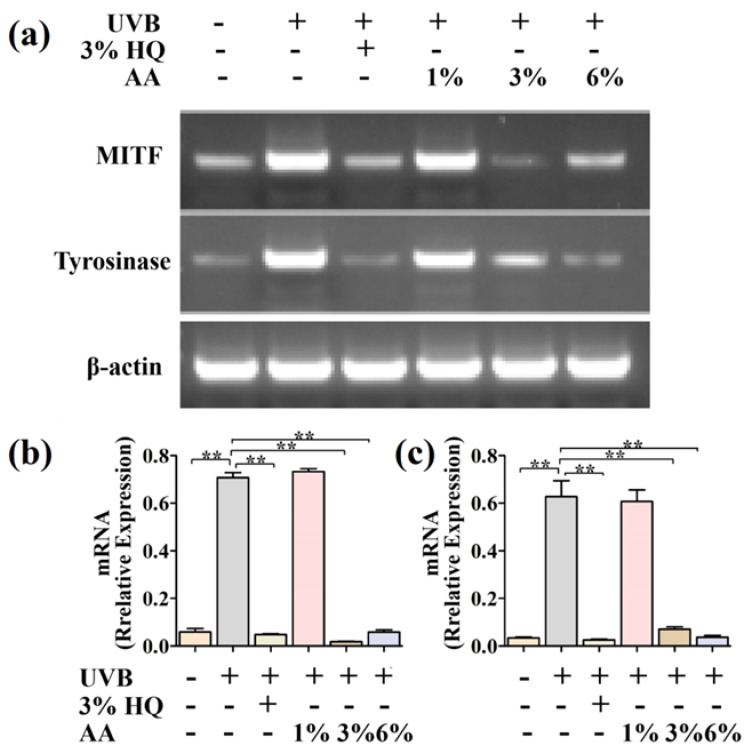
Effects of AA on the expression of mRNA of MITF and tyrosinase of dorsal skin (**a**). The mRNA levels of MITF (**b**) and tyrosinase (**c**) proteins were quantified by densitometry. Data are expressed as mean ± SEM. ** *p* < 0.01 (*n* = 6 per group).

**Figure 8 ijms-23-15952-f008:**
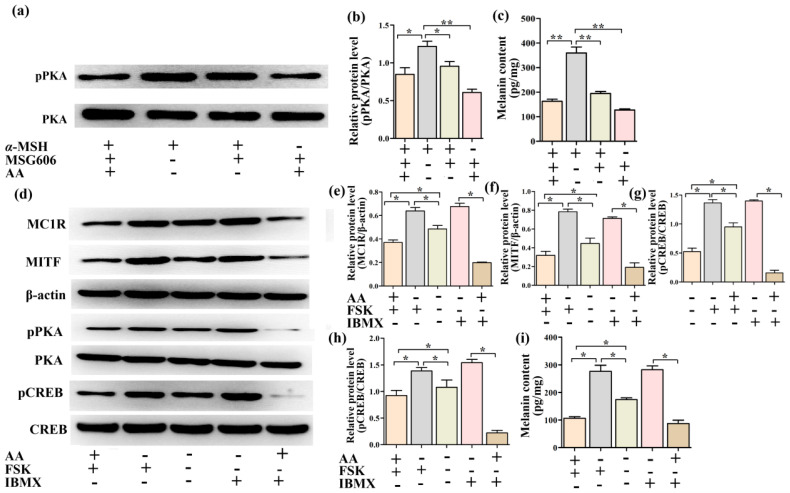
Effects of AA and MSG606 9 (MC1R inhibitor) on the protein expression levels of pPKA (**a**–**c**) and melanin content (**d**) in α-MSH-induced PIG1 cells for 24 h. Effects of AA on the protein expression levels of MC1R, MITF, pPKA and pCREB (**e**–**h**) and melanin content (**i**) in FSK- or IBMX-induced PIG1 cells for 24 h.Data are expressed as mean ± SEM. * *p* < 0.05, ** *p* < 0.01 (*n* = 3 per group).

**Table 1 ijms-23-15952-t001:** The binding energy values between AA and different proteins.

Protein	PDBID	Binding Energy (Kcal/mol)
CREB	7KF0	−23.05
AKT	7D85	−1.53
PKA	4YKO	−12.48
MAPK	6XQ9	−0.92
PI3K	7CIO	−1.53
ERK	4QTA	−1.08
JNK	6LOT	−0.97

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
