# Peer review of "Atraric Acid Ameliorates Hyperpigmentation through the Downregulation of the PKA/CREB/MITF Signaling Pathway"

_ijms, 2022, doi:10.3390/ijms232415952_

Round 1

Reviewer 1 Report

The manuscript titled “Atraric acid Ameliorates Hyperpigmentation through the 2 down-regulation of MC1R/PKA/CREB Signaling Pathway” evaluates the effects of atraric acid on melanin production and investigates which pathways can be responsible for these effects. In particular, Authors aims to demonstrate if AA can inhibit hyperpigmentation through inhibition of tyrosinase activity in α-MSH-213 induced PIG1 cells and in UVB-induced guinea pigs.

These effects of Atraric acids have not been explored and results are of interest, but, at present, there are many experimental drawbacks.

Authors should consider the following points:

1)    Figure 2 panel C: images do not seem to fully support Authors’ statements (i.e. AA reduces the content of melanosomes). Therefore, I recommend providing a quantitation of fluorescence (green) in the different experimental conditions in relation to the number of cells (blue). Size and resolution of EM images are too low to verify the correct interpretation of images. Consequently, quantitation of fluorescence is important. Authors state that cell fluorescence was measured using Image J software (line 314); why Authors do not show the results?

2)    Figure 5: I do not understand why TEM images are taken in the dermis, since melanocytes are present in the epidermal layers. I do not agree with Authors’ interpretation of EM images; cells look more likely to fibroblasts. Again, the quality of the images is very poor, and resolution is not adequate for TEM images. Labels of panels should be explained in the legend.

3)    Materials and methods (4.4): PIG1 cells were treated with Atraric acid dissolved in DMSO. The amount of DMSO is not provided, moreover, since DMSO has a degree of cytotoxicity, the effects of DMSO alone must be investigated and data provided for cells untreated, treated with DMSO and treated with DMSO + AA or other compounds

4)    Materials and methods (4.4): Authors should explain why they are using pancreatin (a mix of different proteolytic enzymes) during the phase of cell adhesion and should provide also the concentration used.

5)    Materials and Methods (4.5 and 4.7): I believe that MA should be replaced by AA, please check

6)    Materials and Methods (4.5): Authors state that “UV irradiation conditions were the same as previously reported with slight differences” and refer to paper #26.  When referring to a paper, this paper should be readable by everybody (I do not know how many readers are familiar with the Korean language and this is the only format that can be found online and at the Journal website). Therefore, I would suggest providing details for instance on the exposure time and intensity. Experiment conducted for 8 weeks is too general. Moreover, there are no information concerning the treatment, beside the dose (v/v). How compounds were administered (all in cream composition? In liquid solution?), how frequent was repeated the treatment? Every time before UV exposure? How was diluted Atraric acid? Methods should provide sufficient details not only to allow experiments to be reproduced in other laboratories, but to verify if the experimental design was appropriate and results are of any significance. At present this evaluation cannot be done.

7)    Materials and methods (4.6): It is not clear how was tyrosinase activity measured?  By Elisa can be easily detect the protein expression (i.e., melanin content), but it is not so clear how was measured an enzymatic activity (since it not a protein measurement). Please provide details.

8)    Supplementary figures: original figures means that images of the whole WB (before any cut and paste) should be provided. Please provide original WB

9)    Check that all abbreviations are described when used for the first time

10) Check for several typing/grammatical errors that are present in the manuscript

Reviewer 2 Report

Title: Atraric acid ameliorates hyperpigmentation through the down-regulation of MC1R/PKA/CREB signaling pathway

             This study indicates that atraric acid (AA) decreased melanin synthesis and tyrosinase in alpha-MSH-activated PIG1 cells, inhibited the phosphorylation of PKA and CREB, as well as UVB-induced hyperpigmentation in guinea pigs. As a mechanism, AA down-regulated MC1R/PKA/CREB signaling pathway.

Major

1)      Experimental data to confirm the molecular simulation, direct binding of AA to both PKA and CREB, may be requisite.

2)      Docking sites of AA on PKA and CREB seem to be related the phosphorylation of PKA and CREB?

3)      The axis is PKA-CREB-MITF-MC1R or tyrosinase? To understand this hypothesis, additional experiment may be requisite to document that AA can inhibit forskolin- or IBMX-induced melanin production in PIG1 cells?

Round 2

Reviewer 1 Report

The revised version of the manuscript by Jing Lee still has some critical aspects that were not (adequately) considered by Authors:

1)    Line 13: Check the dosage of AA. I doubt it is 250 M.

2)    Line 33: misspelled word: melanogenicproteins

3)    Line 58: I suggest to rephrase the sentence: ..attempt to derive melanogenesis inhibitors derived from natural sources

4)    Line 83: Authors are NOT measuring tyrosinase activity. In the response to reviewers Authors attached the sheet of the ELISA kit they used. Once more, it is absolutely clear that Authors are only measuring the amount of tyrosinase and not the enzyme activity. Amount of protein and enzyme activity are two different parameters. You can have a protein but without enzymatic activity (by using inhibitors for instance). The sheet of the ELISA kit never mentions the enzyme activity, since using anti-tyrosinase antibodies, the assay quantitates only the amount of protein. By contrast, there are several tyrosinase assay kits that measure, by colorimetric methods, the ability of tyrosinase to catalyze the conversion of a phenolic substrate to a Quinone intermediate, which reacts forming a highly stable chromophore with absorbance at 510 nm. Therefore, Authors cannot state in the text that they are measuring tyrosinase activity and figure 2c is not correct and referring to tyrosinase activity in the text is not correct. Authors should measure the activity of the enzyme or remove  the word activity in the text, but of course the significance of results is different.

5)    Figure 2:  A new graph was added representing the fluorescence intensity of the images below (and perhaps also additional images). The label on the y axis is not correct: “a-MSH Relative (fluorescence intensity)”. At least parentheses should be removed. More importantly, however, data reported in the graph are not clear.  Compared to controls, a-MSH increased fluorescence intensity that remained unchanged upon arbutin and AA treatments. Where is the inhibitory activity of AA? This was exactly my concern looking at the images. Authors did not respond to  this point.

6)    Line 119: VGLN132: please check

7)    Line 122: SER?? (specify the location of SER residue)

8)    Author’s reply to comment on figure 5 is not clear. Firstly, Authors refer to an error occurring when they refer to ultrastructure images (SEM instead of TEM). Actually, SEM (Scanning Electron Microscopy) were never shown, neither in the first version of the manuscript neither in the revised version. As still written in figure legend, TEM (Transmission Electron Microscopy) images are shown.  Authors explanation “since the resolution of SEM is not high, HE images can also provide strong evidence for the experimental results” is not correct.  Providing that we are dealing with TEM images, the resolution of ultrastructural images is always higher and better compared to H&E histology figures. According to the size of images and the number of pixels of the images, it is difficult even to evaluate if  the labelled organelles are true melanosomes. Again, I disagree with the interpretation of TEM images. Authors are not showing melanocytes as expected, especially looking at the areas boxed in the panels on the left side of the images. Melanocytes are surrounded by keratinocytes, since they are in the epidermal layer of the skin, and they are not between dermal collagen bundles, as shown in the figure. Even cell morphology is different. Moreover, the figure provided in the response to reviewer is different from the image in the manuscript.  I suggest to remove TEM images and I recommend Authors to check that panels at higher magnification of H&E sections are actually the enlargement of the same area in the box of  panels on the right.

9)    Line 175: inflammatory factors infiltrate the skin surface. Which are the inflammatory factors? Authors mean inflammatory cells? Provide image or add “data not shown”. In any case check if the word “factors” is appropriate.

10) Line 179: “pathological sections of the liver and kidney were abnormal..”. In which sense were abnormal? Data in suppl. figure do not seem to support this statement. Moreover, as a general comment, morphology figures are always very small. Therefore, it is mandatory that the image resolution (pixel per unit area) allows readers to enlarge the images on their computer to appreciate details.

11) Figure 7. If image represents a Northern blot detecting mRNA levels, Authors should not refer to “proteins” (line 200) in the figure legend. mRNA expression is not protein expression.

12) Line 216: Figure 8d-h. Panel labelled with these letters are not present. Moreover, figure legend refers to a number of panels and to data that are not provided.

13) Line 254: Add the appropriate reference corresponding to Lee et al.

14) Line 256: Add the appropriate reference to Chu et al.

15) Line 297: is the 10M concentration of DMSO correct? What is the meaning of “mother liquor”? Is the stock solution? Which is the desired concentration of DMSO? Please specify. The concentration is important to exclude cytotoxic effects. This question was raised in the first revision, addressed by Authors in their response, but not adequately revised in the revised version of the manuscript.

16) Line 323: More data are provided concerning UVB exposure, However, I asked to put a reference understandable to all readers. Still, Authors maintained reference 26 (it is not an article in English except for the abstract), in contrast to statements in the response to reviewer.

Reference 28 is not appropriate in this context, since it does not refer to UVB exposure treatments. Authors should be aware that changes reported in the response to Reviewers, should be reported also in the revised text, otherwise revision is useless, and it is not fair and respectful for the time reviewers spent to help the Authors.

17) Line 352: reference 29 is not present in the reference list. Authors should check the entire reference list and the correspondence to references in the text.

18) The English language still requires extensive revision

Author Response

I am very sorry for the inconvenience that you mentioned that I did not modify the original text. I don't know if there is a problem with the uploading process. In fact, I have made serious revisions to the article and marked the revised part with a yellow highlight. Once again express my infinite respect and gratitude to the reviewers! Through these two reviews, I think you are an excellent friend, more like a mentor than a friend. My heart is full of thanks and admiration for you! And thanks you for your significantly meaningful comments on this manuscript. Add We have tried my best to solve the problem you raised, and we would like to report to you, point by point.We appreciate your elaborate efforts in reviewing. Once again, thank you very much! We hope to get your recognition.

Reviewer 2 Report

The manuscript may be reconsidered after major revision.

 Major

1)      Additional experiments should be required to addressing “direct binding of AA to both PKA and CREB” since they are suggested as the molecular targets of AA in the inhibition of hyperpigmentation.

 Minor

1)      The axis may be corrected to PKA/CREB/MITF instead of MC1R/PKA/CREB.

Author Response

To Reviewer: 2

Thanks for your affirmation about our previous studies. Thank you for your significantly meaningful comments on this manuscript, again. We hope to get your recognition.

 Major

 1)   Additional experiments should be required to addressing “direct binding of AA to both PKA and CREB” since they are suggested as the molecular targets of AA in the inhibition of hyperpigmentation.

Answer: We have changed and described the conclusion part, again. There are two findings in our manuscript, the first is the first discovery of the activity of AA to inhibit hyperpigmentation. The second is to reveal the mechanism by which AA inhibits hyperpigmentation by regulating the PKA/CREB/MITF signaling pathway. 

Perhaps the results of the molecular docking section of the article lead you to think that AA acts on the PKA and creb protein targets. But what I need to explain to you here is that the molecular docking results are only results of predicted properties, indicating a possibility. So the conclusion I have drawn here is that the possible targets of AA are PKA and CREB, and the pathway regulated by AA is the PKA/CREB signaling pathway. And in the following experiments, we experimentally verified the signaling pathway. 

In addition, maybe this sentence of “Moreover, co-treatment with α-MSH and AA reduced melanin content and the level of pAKT compared with that in the α-MSH+MSG606 group, which suggested that AA was probably ameliorating the MC1R/PKA/CREB pathway by regulating both MC1R and PKA protein targets.” in the conclusion misunderstood you, I am sorry. "Probably" is used here, which is also an inference. And we have added a sentence of “However the identification of the protein targets of MCIR and PKA requires futher experimental studies.” in the conclusion section of manuscript. In this way, we further express our views that the specific target of AA needs to be proved by follow-up research.

 Besides, I browsed a lot of recent studies on the reduction of melanosis by small molecule compounds in  your journal and found that they also discussed the impact of small molecule compounds on pathways, and did not elaborate on specific targets. The relevant literature is as follows:

  1. Ko, S. C., Lee, S. H. Protocatechuic Aldehyde Inhibits α-MSH-Induced Melanogenesis in B16F10 Melanoma Cells via PKA/CREB-Associated MITF Downregulation. Int. J. Mol. Sci. 2021, 22(8):3861.       
  2. Mun, S. K., Kang, K. Y., Jang, H. Y., Hwang, Y. H., Hong, S. G., Kim, S. J., Cho, H. W., Chang, D. J., Hur, J. S., Yee, S. T. Atraric Acid Exhibits Anti-Inflammatory Effect in Lipopolysaccharide-Stimulated RAW264.7 Cells and Mouse Models. Int. J. Mol. Sci. 2020, 21(19):7070.
  3. Wu, K. C., Hseu, Y. C., Shih, Y. C., Sivakumar, G., Syu, J. T., Chen, G. L., Lu, M. T., Chu, P. C. Calycosin, a Common Dietary Isoflavonoid, Suppresses Melanogenesis through the Downregulation of PKA/CREB and p38 MAPK Signaling Pathways. Int. J. Mol. Sci. 2022, 23(3):1358.

 Minor

2)      The axis may be corrected to PKA/CREB/MITF instead of MC1R/PKA/CREB.

Answer: We agreed with the suggestion of the reviewer. We have changed the title“Alaric acid Ameliorates Hyperpigmentation through the Down-regulation of the MC1R/PKA/CREB Signaling Pathway” to “Alaric acid Ameliorates Hyperpigmentation through the Down-regulation of the PKA/CREB/MITF Signaling Pathway” .

Round 3

Reviewer 1 Report

The manuscript has been revised taking into consideration most of reviewers’ criticisms.

Authors should consider these few points.

1)     Legend to figure 5 (line 180): what is the meaning of histopathological symptoms?  Do you mean histopathological features? Please explain in the legend the acronyms (C, M, LD, MD, HD, 3%HQ) used in the image.

2)     Regarding figure 5, as I already highlighted in my previous comments, TEM images are NOT epithelial cells (nor melanocytes, nor keratinocytes), therefore TEM images (panel b) must be removed. Just leave H&E image.

Author Response

Thanks for your affirmation about our previous studies. Thank you for your significantly meaningful comments on this manuscript, again. We hope to get your recognition.

  • Legend to figure 5 (line 180): what is the meaning of histopathological symptoms?  Do you mean histopathological features? Please explain in the legend the acronyms (C, M, LD, MD, HD, 3%HQ) used in the image.

Answer: We have changed “histopathological symptoms” to “histopathological features”, and changed the legend in the manuscript.

2)     Regarding figure 5, as I already highlighted in my previous comments, TEM images are NOT epithelial cells (nor melanocytes, nor keratinocytes), therefore TEM images (panel b) must be removed. Just leave H&E image.

Answer: We have deleted the TEM images of Figure 2 and Figure 5.

Reviewer 2 Report

---

Author Response

Thanks for your affirmation about our previous studies.
